# Specific Detection and Enumeration of *Burkholderia cepacia* Complex by Flow Cytometry Using a Fluorescence-Labeled Oligonucleotide Probe

**DOI:** 10.3390/microorganisms10061170

**Published:** 2022-06-07

**Authors:** Soumana Daddy Gaoh, Anna Williams, David Le, Ohgew Kweon, Pierre Alusta, Dan A. Buzatu, Youngbeom Ahn

**Affiliations:** 1Division of Microbiology, National Center for Toxicological Research, U.S. Food and Drug Administration, Jefferson, AR 72079, USA; soumana.daddy-gaoh@fda.hhs.gov (S.D.G.); ledavi73@students.rowan.edu (D.L.); oh-gew.kweon@fda.hhs.gov (O.K.); 2Division of Systems Biology, National Center for Toxicological Research, U.S. Food and Drug Administration, Jefferson, AR 72079, USA; anna.williams@fda.hhs.gov (A.W.); pierre.alusta@fda.hhs.gov (P.A.); dan.buzatu@fda.hhs.gov (D.A.B.)

**Keywords:** *Burkholderia cepacia* complex, flow cytometry, fluorescence-labeled oligonucleotide probe

## Abstract

*Burkholderia cepacia* complex (BCC) contamination has resulted in recalls of non-sterile pharmaceutical products. The fast, sensitive, and specific detection of BCC is critical for ensuring the quality and safety of pharmaceutical products. In this study, a rapid flow cytometry-based detection method was developed using a fluorescence-labeled oligonucleotide *Kef* probe that specifically binds a KefB/KefC membrane protein sequence within BCC. Optimal conditions of a 1 nM *Kef* probe concentration at a 60 °C hybridization temperature for 30 min were determined and applied for the flow cytometry assay. The true-positive rate (sensitivity) and true-negative rate (specificity) of the *Kef* probe assay were 90% (18 positive out of 20 BCC species) and 88.9% (16 negative out of 18 non-BCC), respectively. The detection limit for *B. cenocepacia* AU1054 with the *Kef* probe flow cytometry assay in nuclease-free water was 1 CFU/mL. The average cell counts using the *Kef* probe assay from a concentration of 10 μg/mL chlorhexidine gluconate and 50 μg/mL benzalkonium chloride were similar to those of the RAPID-B total plate count (TPC). We demonstrate the potential of *Kef* probe flow cytometry as a more sensitive alternative to culture-based methods for detecting BCC in non-sterilized pharmaceutical raw materials and products with regards to water-based environments.

## 1. Introduction

The *Burkholderia cepacia* complex, otherwise known as BCC, is a group of bacteria comprising 24 different pathogenic species [1]. These species are Gram-negative, catalase-producing, and lactose-non-fermenting [2]. This group has been classified as objectionable microorganisms and opportunistic pathogens that are widespread in the environment, mainly in water and soil [1,2,3,4]. BCC relies on multiple survival mechanisms, including efflux pumps to extrude biocides or render them inactive for growth [2,3,4,5]. This can be observed in their ability to widely contaminate both non-sterile and sterile pharmaceuticals, causing potential life-threatening respiratory infections and serious health risks [6,7,8]. The threats of BCC transmission and spread mainly affect those with underlying and pre-existing health issues, such as individuals who are immunocompromised or have cystic fibrosis or acquired immunodeficiency syndrome [9]. Therefore, it is critical to determine total aerobic bacterial numbers in non-sterile pharmaceutical materials as part of the routine evaluation of microbiological quality [10,11]. Specifically, the detection of very low levels (100 colony-forming units (CFU)/mL) of contaminants is required to ensure acceptable pharmaceutical product quality and safety.

The United States Pharmacopeia (USP) has published a general chapter (USP <60>, Microbiological examination of non-sterile products: tests for Burkholderia cepacia complex) describing the tests needed to ensure that drug ingredients, pharmaceutical water, and finished drug products conform to appropriate quality standards [12]. These tests are mainly limited by their low sensitivity and have been shown to be relatively inaccurate with regards to the detection of BCC [4,13]. To overcome this sensitivity challenge, polymerase chain reaction (PCR), real-time quantitative PCR (qPCR), loop-mediated isothermal amplification (LAMP), recombinase polymerase amplification (RPA), and droplet digital PCR (ddPCR) have also been utilized to test for the presence of BCC [13,14,15,16,17,18,19,20]. PCR has been shown to be rapid, simple, easy to use, and relatively inexpensive for the detection of BCC [4]. LAMP and RPA are efficient isothermal methods that eliminate the need for a thermal cycler and have high deployment potential in resource-limited settings [4,18,19,20]. Two major limitations of qPCR are of particular concern, namely its inability to distinguish live from dead cells and the amplification reactions sensitivity to inhibitors.

One of the most commonly used methods to distinguish live and dead cells in bacterial samples is flow cytometry [21,22,23,24,25]. This method is highly sensitive, extremely precise, and is specifically noted for its ability to detect down to one single viable bacterial cell in a short period of time compared to other available methods. Most recently, flow cytometry has exhibited the highest sensitivity for detecting trace levels of BCC strains [14]. Although flow cytometry demonstrates advantages such as a short assay time and high sensitivity, specificity towards a target organism is necessary in non-sterile pharmaceutical products. Fluorophores can be coupled with flow cytometry to rapidly detect specific bacteria [25,26,27]. This is particularly useful in the case of BCC, as the ability to detect one single cell in a sample would be beneficial to effectively prevent and avoid potential contamination. Alexa Fluor 488 is known for its superior photostability and signal intensity and is stable at lower pH levels [28]. The higher fluorescence intensity of Alexa 488 conjugates is advantageous for applications in which a direct strong signal is needed to detect BCC. However, due to the probe’s length, it drastically reduces specificity towards BCC, thereby limiting its specific detection. While specifically developed for flow cytometry, its use has proven to be technically challenging.

Our aim is to establish a more efficient and cost-effective method to detect BCC in order to prevent widespread contamination. In this study, we developed a rapid-flow-cytometry-based detection method for BCC using an oligonucleotide probe targeting the inner-membrane protein of the KefB/KefC family sequence. After the optimization of the hybridization temperature and probe concentration, we tested the specificity and sensitivity of the developed assay. Finally, we applied the developed probe-based assay for the detection of BCC in antiseptics (i.e., 10 μg/mL chlorhexidine gluconate (CHX) or 50 μg/mL benzalkonium chloride (BZK)) and surface water.

## 2. Materials and Methods

### 2.1. Bacterial Strains and Preparation

The *Burkholderia* strains used in this study are listed in Appendix A. Thirty-six strains of *Burkholderia* spp. and two strains of *Caballeronia* spp. were obtained from the *Burkholderia cepacia* Research Laboratory and Repository at the University of Michigan. All 20 BCC strains and 18 non-BCC strains were cultivated as described previously [14].

To determine BCC in autoclaved nuclease-free water (Qiagen, Valencia, CA, USA) and antiseptics, samples were prepared in autoclaved nuclease-free water (23 January 2020), containing 10 µg/mL CHX (8 September 2020) and 50 µg/mL BZK (8 September 2020), then stored at 23 °C for 524, 295, and 295 consecutive days, as described previously [14,18].

### 2.2. Design of the Kef Probe

The inner-membrane protein of the KefB/KefC family (*Kef*) probe (5′-Alexa488-O-ATGCAGCGGCTCGTGTTCGG-3′) was designed as previously described [18]. The peptide nucleic acid *Kef* probe was conjugated to Alexa Fluor 488 (Eurofins Genomics LLC, Louisville, KY, USA) and reconstituted prior to use in molecular-grade water to a 100 μM stock concentration.

### 2.3. RAPID-B Total Plate Count (TPC)

The TPC reagent used here contained thiazole orange (TO) and propidium iodide (PI), as described previously [23]. Thiazole orange is a cell-permeable dye which intercalates with the DNA of live cells, generating green fluorescence (FL1 channel). Only dead cells with compromised membranes may be penetrated by PI, which quenches the TO and emits red fluorescence (FL3 channel). As a result, combining these two dyes yields a quick approach for total cell quantification as well as distinguishing between live and dead cells (Appendix A). BCC samples were prepared as previously described [14]. In summary, 1 mL of each dilution (10 and 10^2^ CFU/mL) of BCC suspension was centrifuged at 16,300× *g* for 5 min, and the cell pellets were washed twice with 1 mL of 1× phosphate-buffered saline (PBS; Sigma-Aldrich, St. Louis, MO, USA). The pellets (approx. 100 μL) were resuspended in 567 μL PBS, and 333 μL of TPC reagent was added to each suspension. The flow cytometer used was a model A40 flow cytometer (Apogee Flow Systems, Hemel Hempstead, UK).

### 2.4. Optimization of Kef Probe Assay for BCC 

A Synergy MX spectrophotometer (BioTek Instruments, Winooski, VT, USA) was used to adjust the optimal density at 600 nm to around 0.1 (range = 0.095–0.105), which is equivalent to about 10^8^ CFU/mL. Serial dilutions of each BCC suspension were prepared in 4 mL of autoclaved nuclease-free water to yield appropriate CFUs (10, 10^2^, 10^3^, and 10^4^ CFU/mL), as previously described [14,18]. The *Kef* probe assay was performed by a modified method described previously in [25]. Briefly, the samples (10 and 10^2^ CFU/mL) were centrifuged at 16,300× *g* for 5 min, and the supernatant was removed. The cell pellets were washed with 1 mL of 1× PBS and centrifuged again at 16,300× *g* for 5 min. The pellets were resuspended in 1 mL of 10% buffered formalin (Sigma-Aldrich, St. Louis, MO, USA) and incubated at room temperature for 30 min under gentle vortexing on a Vortex-Genie 2 (Daigger, Wheaton, IL, USA). The samples were centrifuged and washed with 1 mL 1× PBS. The pellets were resuspended in 290 μL hybridization buffer (20 mM Tris-HCl (pH 7.5), 5 mM EDTA, 3 M NaCl, and 0.01% SDS) and 10 μL of *Kef* probe and incubated at 60 °C for 30 min. After incubation of BCC, samples were washed with 1 mL hybridization buffer at 60 °C for 45 min with vortexing (1000 rpm). The samples were centrifuged again at 16,300× *g* for 5 min. Pellets were resuspended in 1 mL 1× PBS and used for flow cytometeric analyses. 

Three different concentrations of the *Kef* probe, calculated to be 1, 10, and 100 nM, at temperatures of 55, 60, and 65 °C were tested to determine the optimal hybridization conditions. For control purposes, with or without cells, a blank *Kef* probe at 1, 10, and 100 nM with nuclease-free water was also analyzed. The samples were analyzed on the A40 flow cytometer to determine bacterial concentration [14]. This experiment was performed in triplicate.

### 2.5. True-Negative Rate (Specificity) in Kef Probe Assay

To investigate the specificity of the flow cytometer with the newly designed *Kef* probe, 20 BCC strains and 18 non-BCC strains were analyzed, three times each, using the TPC assay and *Kef* probe assay as described above, and the averages were calculated. Based on the *Kef* probe blank counts, positive results were recorded (0 = not detected, 1 = detected) that confirmed counts above 10^3^ cells/mL (i.e., 100 events/100 µL). The formula used to calculate the estimated specificity is as follows: estimated specificity (%) = number of true-negative events (TN)/(number of false-positive events (FP) + TN) × 100 [29].

### 2.6. Limit of Detection (LOD) in Kef Probe Assay

The LOD was measured using the *Kef* probe assay and by analyzing tenfold serial dilutions (10^3^, 10^2^, 10, and 1 CFU/mL) of *B. cenocepacia* AU1054. A *Kef* probe assay was considered positive when the counts were above 10^3^ cells/mL. 

### 2.7. Comparison of TPC and Kef Probe Assay

#### 2.7.1. BCC-Spiked 10 μg/mL CHX and 50 μg/mL BZK Samples 

Prior to the *Kef* probe assay, the total bacterial population was measured using a TPC assay. For control purposes, a blank *Kef* probe with nuclease-free water and 10 μg/mL CHX and 50 μg/mL BZK blank samples (non-BCC-spiked) were analyzed on the flow cytometer. BCC-spiked 10 μg/mL CHX and 50 μg/mL BZK samples were prepared (8 September 2020) and stored at 23 °C for 295 consecutive days, as previously described [18]. To compare the *Kef* probe assay with the TPC assay, *B. cenocepacia* AU0222, *B. cepacia* PC783, *B. cepacia* AU24442, and *B. stabilis* AU23340 suspensions were prepared as described above using a dilution equivalent to 10 CFU/mL.

#### 2.7.2. Surface Water

Using sterilized 50 mL conical tubes (Eppendorf, Enfield, CT, USA), six water samples were collected in July 2021 from surface waters in Arkansas including: Arkansas River in Jefferson, AR (07/22/21); Creek 1 in Bryant, AR (07/25/21); Creek 2 in Little Rock, AR (07/25/21); Creek 3 in Hot Springs Village, AR (07/25/21); and tap water and cold fountain water in Hot Springs, AR (07/26/21) as drinking water. 

In order to obtain total bacteria counts, 10 μL aliquots of serial dilutions of each water sample were dropped directly onto 1/10 strength tryptic soy agar (1/10 × TSA) and 1/10 × TSA with *Burkholderia cepacia* selective supplement (200 μL of *Burkholderia cepacia* selective supplement containing polymyxin B (150,000 IU/L; 3.75 µg/mL), gentamicin (5 µg/mL), and ticarcillin (100 µg/mL): Oxoid, Lenexa, KS, USA) [30]. The plates were incubated at room temperature (23 °C) for 48 h before enumeration. 

For the flow cytometric determination of BCC from surface water, we counted directly from surface water and from water samples spiked with *B. cenocepacia* HI2976. Prior to the *Kef* probe assay, the total bacterial population was measured using the TPC assay from non-spiked and BCC-spiked samples. For non-spiked water samples, dilutions were prepared to approximate 10 CFU/mL based on the above results. For BCC-spiked samples, 900 µL of the water sample was spiked with 100 µL of *B. cenocepacia* HI2976 (the BCC cell concentration was approximately 1.5 × 10^8^ cells/mL) and exposed to water for 1 h to allow *B. cenocepacia* HI2976 cells to equilibrate with the ambient environmental conditions. The dilutions were prepared to approximately 10 CFU/mL based on non-spiked water samples. The BCC-spiked and non-spiked samples were analyzed using the TPC and *Kef* probe assays, as described above. For negative control purposes, a filtered (0.2 µm pore size, 25 mm in diameter) blank sample was also analyzed. 

## 3. Results

### 3.1. Kef Probe Designed for Flow Cytometry

Using the genome-gene matrix (GGM), we identified 197 BCC-specific fragments (20–40 bp) that are only present in BCC genomes. Among the 197 BCC-specific fragments, we chose fragment 4056 (20 bp) (the inner membrane protein of the KefB/KefC family), which showed 100% sequence identity, and three copies (orthologues) in BCC. This allowed for more binding areas on the cell surface, and thus more fluorescence, which facilitated the identification of BCC in the flow cytometer when counting cells. The *Kef* probes had a 65% G+C content, a melting temperature of 66.6 °C, and a length of 20 bases.

### 3.2. Optimal Concentration and Temperature of Hybridization Kef Probe

Three different concentrations of the *Kef* probe (1, 10, and 100 nM) were tested to find the optimal concentration that would enable the efficient detection of BCC (Figure 1). The three different concentrations of the *Kef* probe at 1, 10, and 100 nM corresponded to 2164.2, 2589.2, and 2151.7 cells/mL for every 10 CFU/mL of *B. cenocepacia* AU1054, respectively (Figure 1a). However, the main difference was observed with blanks at concentrations of 1 and 10 nM, which had the lowest number of cells/mL (280.8 and 670.8). A concentration of 100 nM showed a high background fluorescence with 1432.5 cells/mL, which could potentially yield false-positive results. Based on observations of the blanks, values above 10^3^ cells/mL (i.e., 10^2^ events/100 μL) were considered positive. A minimum of 10^2^ events/100 μL for each sample were counted, suggesting that the 1 nM concentration was optimal, as a high number of events on a blank would be undesirable. 

To find the optimal temperature for the *Kef* probes, three temperatures (55, 60, and 65 °C) were tested and compared with the TPC results (Figure 1b). The optimal temperature for the *Kef* probe hybridization is the temperature that allows for the counts to be closest to those of the TPC assay. The TPC assay yielded counts of 660–1110 cells/mL. The *Kef* probe assay counts were obviously different below 55 °C (1340–1660 cells/mL, i.e., more than 10^2^ events/100 μL) and above 65 °C (110–540 cells/mL) (*p* < 0.05). The *Kef* probe assay at 60 °C (820–1120 cells/mL), being the closest to the TPC, was selected to be the optimal temperature for the use of the *Kef* probe.

### 3.3. True-Positive Rate (Sensitivity) and True-Negative Rate (Specificity)

To investigate the specificity of flow cytometry with the newly designed *Kef* probe, 20 BCC strains and 18 non-BCC strains were subjected to the probe assay. The 38 *Burkholderia* strains were individually serially diluted, and about 10^2^ CFU/mL was used for the hybridization reaction. As shown in Figure 2a, the 20 BCC strain numbers determined by the TPC assay averaged 2999.3 cells/mL, and an average of 1842.9 cells/mL when the cell number estimates were based on the *Kef* probe. The *Kef* probe counted over 10^3^ cells/mL for 18 BCC strains. The average cell numbers of *B. contaminans* HI3429 (793.3 cells/mL; ranging from 810 to 1190 cells/mL) and *B. diffusa* AU1075 (902.2 cells/mL; ranging from 790 to 1660 cells/mL) were below 10^3^ cells/mL. Positive results were recorded (0 = not detected, 1 = detected) that confirmed counts above 10^3^ cells/mL. The sensitivity of the *Kef* probe was 90% ((number of true-positive events (TP)/(TP + number of false-negative events (FN)) × 100 = 18/(18 + 2) × 100 = 90% sensitivity; 18 positives out of 20) (Figure 2b). 

The cell concentrations of 18 non-BCC strains were determined by TPC, averaging 1934.5 cells/mL, whereas an average of 349.9 cells/mL was obtained by the *Kef* probe assay (Figure 2c). The *Kef* probe counted less than 10^3^ cells/mL for 16 non-BCC strains. The counts for *B. fungorum* AU35949 (1063.3 cells/mL) and *Caballeronia zhejiangensis* AU12096 (1035.6 cells/mL) were above 10^3^ cells/mL. Based on these results, the specificity of the *Kef* probe was 88.9% ((number of true-negative events (TN)/(number of false-positive events (FP) + TN) × 100 = 16/(2 + 16) × 100 = 88.9%; 16 negatives out of 18) (Figure 2d). 

### 3.4. Limit of Detection (LOD) of the Flow Cytometry Assay 

To determine the LOD of the flow cytometry assay, we tested the *Kef* probe using diluted solutions of *B. cenocepacia* AU1054 (Figure 3). The counts for *B. cenocepacia* AU1054 (1, 10, 10^2^, and 10^3^ CFU/mL) were determined by TPC, averaging 2003.3 cells/mL, 2082.2 cells/mL, 3627.8 cells/mL, and 22397.8 cells/mL, respectively. When cell number estimates were solely based on the *Kef* probe assay, the abundances (average: n = 9) of *B. cenocepacia* AU1054 (1, 10, 10^2^, and 10^3^ CFU/mL) averaged 1438.9, 2244.4, 3358.9, and 4041.1 cells/mL, respectively. Analysis by probe clearly detected 1 CFU/mL, which is equivalent to 1438.9 cells/mL (143.8 events/100 μL) in a test reaction and appeared to be strongly associated with CFU/mL (r^2^ = 0.7484; y = 1.6552x + 173.5) (Figure 3). 

### 3.5. Counting BCC from Antiseptic Solutions and Surface Water with the Kef Probe Assay 

#### 3.5.1. Long-Term Storage in 10 μg/mL CHX and 50 μg/mL BZK 

A comparison of the sensitivity of the TPC and *Kef* probe assay used to detect BCC strains from 10 μg/mL CHX and 50 μg/mL BZK kept at 25 °C for 295 days is presented in Figure 4. The average *Kef* probe assay at a concentration of 10 μg/mL CHX averaged 1665.0 cells/mL, for 10 CFU/mL (Figure 4a). Similarly, the average cell count using the TPC assay averaged 1694.9 cells/mL. However, the TPC assay for *B. stabilis* AU23340 averaged 697.8 cells/mL, whereas the *Kef* probe cell count averaged 2588.3 cells/mL (Figure 4b).

At a concentration of 50 μg/mL BZK, the TPC and *Kef* probe assay were similar, yielding counts of 1993.9 cells/mL and 1761.0 cells/mL, respectively (Figure 4c). However, the count for *B. stabilis* AU23340 quantified by TPC averaged 3966.7 cells/mL, whereas an average of 966.7 cells/mL was obtained by the *Kef* probe assay (Figure 4d).

#### 3.5.2. Surface Water

Six surface water samples (four non-drinking water and two drinking water samples) were collected and analyzed to determine whether the *Kef* probe can distinguish BCC from other microbial populations commonly present in water sources. We counted a total of 10^3^~10^5^ CFU/mL on 1/10× TSA inoculated with each of the six surface water samples (data not shown). The plate cultures could not enumerate BCC on 1/10× TSA with *Burkholderia cepacia* selective supplement. Consequently, we spiked the water samples with BCC prior to flow cytometry analyses.

BCC-spiked water samples were individually serially diluted, and about 10 CFU/mL was used for flow cytometry. The average results for the TPC cell counts ranged from 1892.2 to 6271.1 cells/mL for the six different samples (Figure 5). The *Kef* probe cell counts for the water samples (non-spiked with BCC) did not show any BCC (data not shown), but the *Kef* probe cell counts for the BCC-spiked samples yielded average counts ranging from 110 to 1546 cells/mL. As shown in Figure 5, samples collected from the Arkansas River and drinking water (tap water and fountain water) had average counts ranging between 1057.1 and 1546.6 cells/mL. Less than 10^3^ cells/mL BCC were detected in Creek 1 (778.9 cells/mL), Creek 2 (110), and Creek 3 (196.7 cells/mL) using the *Kef* probe assay in samples spiked with BCC. 

## 4. Discussion

The BCC contamination of unsterilized pharmaceutical products is a major concern and has been responsible for multiple recalls in the past few years. The *Kef*-probe-based flow cytometric assay for the specific detection of BCC—as described here—can detect 1 CFU/mL at a concentration of 1 nM *Kef* probe at 60 °C. We demonstrated that the *Kef* probe assay, using flow cytometry, was able to detect BCC in antiseptics after 295 days and in drinking water after the optimization of the *Kef* probe.

The flow-cytometry-based system, RAPID-B, is a well-established method for counting and assessing the viability of bacteria in real time [14,21,22,23,24]. To analyze mixed microbiota, RAPID-B genetic-based assays have been developed, resulting in an assay that can be run at a much lower cost and over a shorter timeframe, with a high sensitivity and specificity [25]. Using a 5′-Alexa Fluor 488 dye-conjugated oligonucleotide probe, we can detect a small number of *Escherichia coli* cells in the presence of large numbers of other bacteria, because a single *E. coli* cell harbors many copies of 5S, 16S-like, and 23S-like rRNAs in 10^5^ ribosomes [26]. The more orthologues there are, the more target regions for the hybridization of the probes when using flow cytometry. Indeed, individual cells can be identified by radiolabeled oligonucleotides due to the high number of copies [31]. In this study, increased fluorescence allowed BCC to be identified due to three orthologue copies and their makeup having 100% sequence identity, which the probe was designed for. Our experiments demonstrated that the *Kef* probe was optimal for detecting BCC in conjunction with flow cytometry. Here, 90% sensitivity (i.e., the number of true-positive events) was demonstrated using flow cytometry. Among the 20 BCC strains, the cell numbers of *B. contaminans* HI3429 and *B. diffusa* AU1075 ranged from 790 to 1660 cells/mL, which is considered positive. Since a cell count of 280.8 cells/mL was observed with blanks at concentrations of 1 nM, we used a cutoff value of over 500 cells/mL instead of 10^3^. In addition, if at least one out of five replicates had a count over 10^3^ cells/mL, the event was labeled “TP”. Even though these counts were close to 10^3^ cells/mL, the failure to detect these strains may have been due to a lack of cell permeabilization during the hybridization process, or more likely the low efficiency of the probe binding to the target [26]. Another possible explanation is that the physiological status of cells is directly correlated to signal intensity, which is not measured [32]. Conversely, among the 20 non-BCC strains, *B. fungorum* AU35949 and *Caballeronia zhejiangensis* AU12096 resulted in cell count averages above 10^3^ cell/mL, which is indicative of false-positive events. These results may be explained by the presence of an unspecified binding site within these two strains [27]. No such sequences were found by BLAST analysis in the two non-BCC strains that tested positive with the *Kef* probe. The relatively high sensitivity and specificity of the *Kef* probe makes it an appropriate choice for the flow cytometric identification and enumeration of BCC.

In the present study, different concentrations of the probe and varying temperatures were tested to determine the optimal conditions for the assay. Xue et al. [25] used a 220 nM DNA probe (from a stock solution of 200 ng/μL) and an incubation temperature of 55 °C for 30 min. Joachimsthal and co-workers added a fluorescently labeled probe at a rate of 100 pmol/mL (100 nM) at 40 or 46 °C for 30 min [27]. In preparation for flow cytometry analysis, the final cell concentrations were approximately 10^6^–10^9^ cells/mL. The highest sensitivity for detecting over 10^6^ cells/mL requires higher probe concentrations [25,27]. In order to keep the background fluorescence low, it is beneficial to use the minimum concentration of the *Kef* probe required to detect BCC cells ranging from 1 to 10^3^ CFU/mL. We tested three concentrations (1, 10, and 100 nM), which yielded comparable event counts per milliliter at a concentration of 10 CFU/mL. The use of the *Kef* probe at a concentration of 1 nM is considered optimal, since it is sufficient to hybridize 10^3^ CFU/mL of BCC strains. Furthermore, for control purposes, *Kef* probe blanks with only CHX and BZK added were also analyzed on the flow cytometer; they showed 265 and 216 events/100 μL, respectively, which are considered positive results. These background fluorescence results were similar to our previously obtained TPC assay results [14]. In order to reduce false positive results for the *Kef* probe assay, CHX and BZK (negative control) samples were washed in 1× PBS, similar to the treated or spiked samples. False-positive counts (at approximately 300 events/100 μL) were discarded by adjusting the cutoff value. Consequently, the lowest probe concentration yielded low background fluorescence, sufficient to hybridize 10^3^ CFU/mL of BCC strains. 

BCC in nuclease-free water, as well as BCC in CHX and BZK solutions, was previously detected down to 1 CFU/mL by the flow cytometric method using a TPC assay [14]. This LOD is ideal and provides a decent detection strategy for pharmaceutical products where BCC contamination is possible. Low event counts in pharmaceutical products often evade detection; consequently, a trace-level detection technology is essential. The current study confirms that the *Kef* probe assay can detect BCC specifically down to 1 CFU/mL. The assay’s useful range is from 1 to 10^3^ CFU/mL. The results show that the reaction saturation for the *Kef* probe was reached at a concentration of 10^4^ CFU/mL. Therefore, it was impossible to quantify the number of counts at this concentration. For instance, with 10^4^ CFU/mL *B. cenocepacia* AU1054, the TPC averaged 22,397.8 ± 3308.8 cells/mL, while the *Kef* probe cell counts averaged 4041.1 ± 345.3 cells/mL. The cell number according to the TPC assay should have been similar to the *Kef* probe assay, but they showed a ten-fold difference (Figure 3). Concurrently, with *B. cenocepacia* AU1054 cell concentrations below 10^3^ CFU/mL, the cell counts observed by the TPC and *Kef* probe assays remain comparable. These results indicate that at higher cell densities, our 1 nM *Kef* probe concentration may be limited to accurately quantifying higher BCC cell numbers. However, based on the fact that the detection of BCC in pharmaceutical products requires a trace-level detection capability, the working range of the *Kef* probe assay from 10^0^ to 10^3^ CFU/mL is very applicable. It is also undesirable to increase the probe concentration in order to pick up higher cell concentrations of BCC, since the fluorescence background would increase, which could produce false-positive results. To overcome this problem, ensuring the use of under 10^3^ CFU/mL with the *Kef* probe at a concentration of 1 nM will allow for accurate BCC detection.

Flow cytometry is a very sensitive method allowing for high-precision detection, at the level of one viable bacterial cell, in a short period of time [14,21,22,23,24,25]. Furthermore, previous work by our co-authors has shown that the TPC assay containing DNA dyes can stain bacterial cells to assess whether they are alive, injured, or dead [21]; this was applied to assess the effectiveness of different antiseptics on BCC [14]. However, these tests may be limited by their specificity [25,26,27]. The TPC assay is a generic live/dead assay for all bacteria and lacks specificity. Previous studies also demonstrated that the flow cytometry genetic probe assays can be pathogen-specific and detect trace levels of the pathogen [25,27]. In this study, we demonstrated the similar sensitivity (true-positive rate) of the TPC and *Kef* probe assays for detecting BCC at 10 CFU/mL in CHX and BZK solutions. The only exception was *B. stabilis* AU23340 (Figure 4). In CHX, with *B. stabilis* AU23340, the *Kef* probe assay yielded a much higher event count compared to the TPC assay. A possible explanation for this result could be the incomplete washing out of the non-specifically bound *Kef* probe. As shown in our previous work, multiple washing reduced background fluorescence [14]. Conversely, the *Kef* probe assay in BZK solution with *B. stabilis* AU23340 yielded a much lower event count compared to the TPC assay. This may be explained by the less-than-optimum hybridization conditions for the *Kef* probe to bind *B. stabilis*, as theoretically we should have seen binding similar to that with BCC. Overall, although the CHX and BZK (negative control) samples needed to be washed in PBS to reduce the presence of random particulate matter, similar results for the *Kef* probe and TPC assays were obtained in both CHX and BZK samples.

Tests on surface water were also performed in order to determine whether the *Kef* probe assay can distinguish BCC from other naturally occurring bacteria, since multiple microbial populations are commonly present in pharmaceutical-grade water. Furthermore, non-drinking surface water comprises high concentrations of interfering particles such as bacteria and organic as well as inorganic particles [33,34]. Our surface water samples did yield different results depending on the amount of organic and inorganic particles present. BCC-spiked drinking water (tap water and cold fountain water) showed average counts reaching over 10^3^ cells/mL of bacteria for the TPC and *Kef* probe assays, which is a positive result. However, Creeks 2 and 3 presented a relatively low amount of detected BCC (less than 10^3^ cells/mL) compared to other samples (Figure 5). Indeed, this may be due to the presence of many interfering particles in Creeks 2 (2696.7.0 ± 140.1) and 3 (2970.0 ± 132.3) compared to the other filtered blank samples. This shows that the presence of higher concentrations of particulate matter can negatively affect enumeration. Conversely, the lower amounts of particles in water samples such as in drinking water will increase the accuracy of BCC detection. 

While using flow cytometry to analyze water samples remains challenging, a few general concerns are worth mentioning. First, BCC-specific probe design is one of the main limiting factors, due to the close genomic and phenomic relatedness between BCC and other *Burkholderia* strains. Readily available complete genome sequence data of *Burkholderia* will ensure more specific probe design (i.e., further sequence surveys will lead to improved probe design). Despite the fact that our probe design covers all BCC strains, we were unable to test two major strains encountered in cystic fibrosis patients, namely *B. multivorans*, and *B. vietnamiensis*, due to a stock shortage. Furthermore, the relatively small size of the oligonucleotide hybridization probes minimizes the problems of cellular permeability and access to binding sites. Secondly, even though flow cytometry is a highly sensitive method, the enumeration of rare events (i.e., 1 targeted cell overshadowed by 1,000,000 non-targeted cells) is extremely difficult [35]. The *Kef* probe assay, intended for BCC detection in environmental water samples, should be adjusted in terms of the cell number with regard to the sample matrix. The pre-enrichment of water samples in 1/10× TSB prior to flow cytometry analyses seems useful in recovering BCC cells that may be in a debilitated state [30,36,37]. Thirdly, using flow cytometry to analyze dirty water samples remains challenging and requires the mechanical or optical separation of cells from interfering particles [33,34]. For this problem, it has been previously demonstrated that centrifugation using a gradient medium such as Percoll^®^ allows the cleanup of dirty matrices [22]. It is possible that this method could be used in future studies to significantly reduce background fluorescence due to particulate matter. Fourthly, the more wash steps the samples undergo, the less accurate and absolute the event counts will be. Consequently, to minimize error margins, it is recommended for samples to be examined repeatedly. Specifically, antiseptics such as CHX and BZK (negative control) samples need to be washed, since they exhibit autofluorescence. In the event of high background fluorescence emanating from the matrix, samples may be subject to photobleaching. Lastly, 10% buffered formalin kills surviving bacteria, suggesting that the *Kef* probe assay is unable to discern dead from live cells, as seen with the TPC assay. Further studies are required to evaluate alternative approaches.

## 5. Conclusions

BCC continues to be a major issue in pharmaceutical product contamination, and therefore better detection methods than those currently available are in high demand. This study showed that the combination of a fluorescent-labeled *Kef* probe, designed from a housekeeping gene, with flow cytometry is a powerful tool to provide specific, sensitive, and accurate BCC detection in water-based environments. The *Kef* probe-based flow cytometry assay described here demonstrated its effectiveness with its ability to detect 1 CFU/mL of BCC. The ability to detect BCC in various solutions, including the two antiseptics CHX and BZK, has implications for the assay’s potential. Furthermore, the assay was optimized to be simple and cost-effective to ensure reproducibility. Due to growing concerns for possible BCC contamination and other rising health issues, a detection assay such as that used in this study has the potential to improve public health. 

## Figures and Tables

**Figure 1 microorganisms-10-01170-f001:**
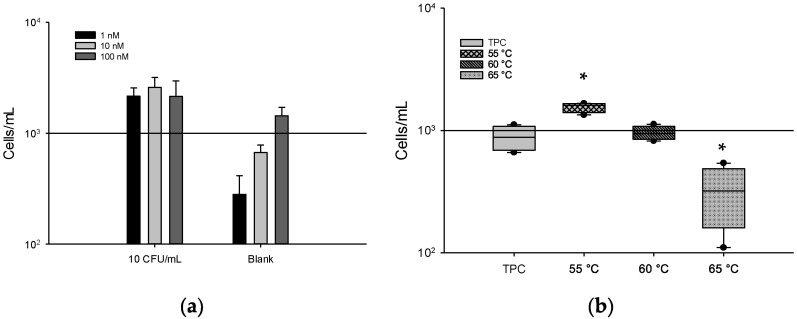
Different concentrations and temperature conditions of hybridization using 10 CFU/mL of *B. ambifaria* HI2468. (**a**) *Kef* probe concentrations of 1, 10, and 100 nM; (**b**) hybridization temperatures of 55, 60, and 65 °C. Horizontal lines show median values, boxes denote values within the lower and upper quartiles of the data. * Indicates statistically significant differences from TPC (*p* < 0.05). Statistical analysis was performed via one-way analysis of variance (ANOVA) and Tukey test using SigmaPlot vs. 13.0 software (Palo Alto, CA, USA).

**Figure 2 microorganisms-10-01170-f002:**
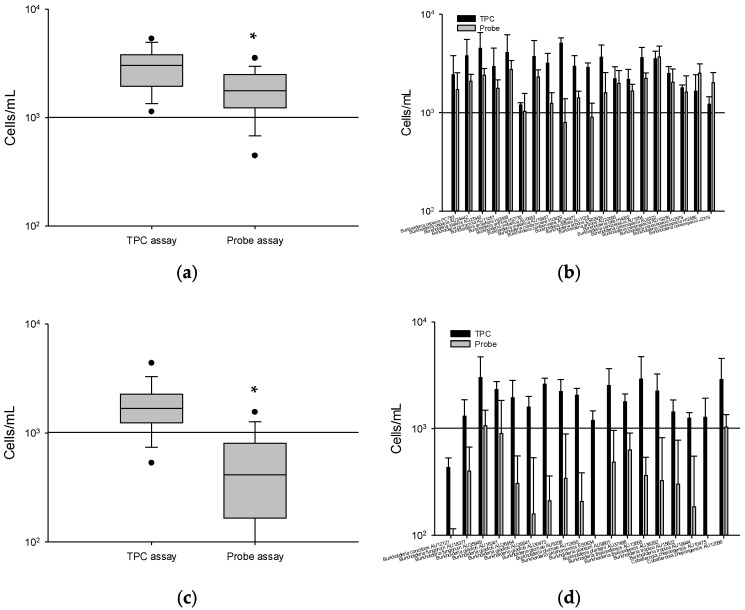
Comparison of specificity of *Kef* probe for 100 CFU/mL of 20 BCC strains (**a**,**b**) and 18 non-BCC strains (**c**,**d**) using flow cytometry. * Indicates statistically significant differences from TPC (*p* < 0.001). Statistical analysis was performed via a paired t-test using SigmaPlot vs. 13.0 software. See Figure 1 for legends.

**Figure 3 microorganisms-10-01170-f003:**
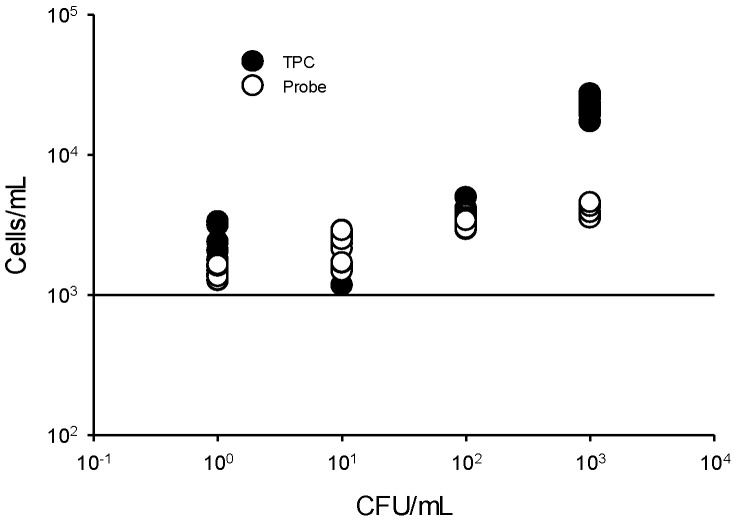
Limit of detection (LOD) of flow cytometry using 10-fold serial dilutions of *B. cenocepacia* AU1054 ranging from 1 to 10^3^ CFU/mL.

**Figure 4 microorganisms-10-01170-f004:**
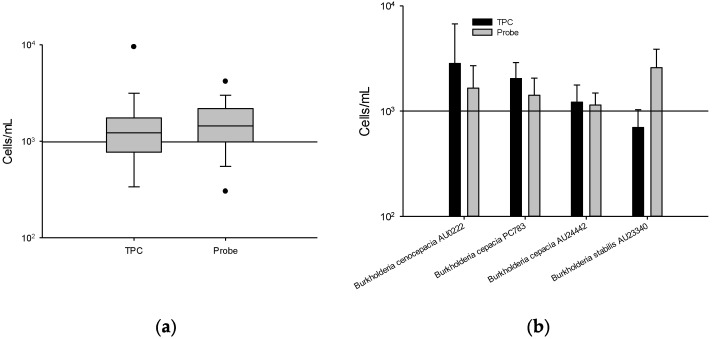
Bacterial abundance assessed for 10 CFU/mL by the TPC and *Kef* probe assays using flow cytometry in CHX (**a**,**b**) and BZK (**c**,**d**). There were no significant differences between probe and TPC (**a**: *p* = 0.329, **c**: *p* = 0.981). Statistical analysis was performed via a paired t-test using SigmaPlot vs. 13.0 software.

**Figure 5 microorganisms-10-01170-f005:**
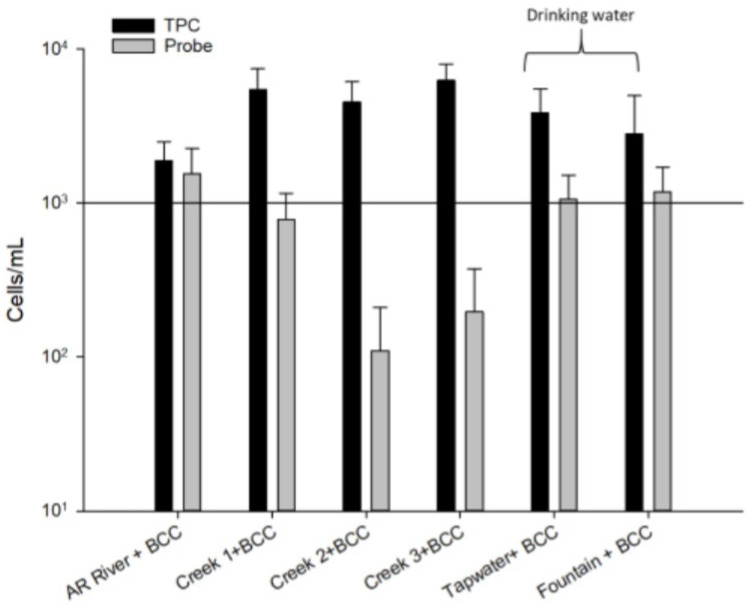
Bacterial abundance assessed in several different surface water samples; spiked with *B. cenocepacia* HI2976 (10 CFU/mL) using the TPC and *Kef* probe assays.

## Data Availability

The data presented in this study are available on request from the corresponding author.

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
