# Peer review of "Specific Detection and Enumeration of Burkholderia cepacia Complex by Flow Cytometry Using a Fluorescence-Labeled Oligonucleotide Probe"

_microorganisms, 2022, doi:10.3390/microorganisms10061170_

Round 1

Reviewer 1 Report

The authors state in line 41 that “The threats of BCC transmission and spread are mainly for those individuals with underlying and pre-existing health issues, such as people who are immunocompromised, have cystic fibrosis, acquired immunodeficiency syndrome, or other health issues”. For many laboratories, BCC are most frequently encountered in patients with CF and, in this context. B. multivorans is the dominant pathogen accounting for up to half of all BCC infections in some studies (e.g. https://pubmed.ncbi.nlm.nih.gov/32946281/). It is therefore surprising that B. multivorans has not been included among the strains of BCC that were tested (Table S2). To a lesser extent, B. vietnamiensis is a surprising omission. Neither of these species are mentioned in the entire paper. It should be emphasised as a limitation that not all species among the BCC have been tested and I would suggest making specific reference to B. multivorans.

Line 52 states that: “These tests are mainly limited by their low sensitivity and have been shown to be relatively inaccurate with regards to detection of BCC”. Can the authors provide a reference to back up this claim about inaccurate detection of BCC?

Line 62: “….sufficient sensitivity of amplification reactions to inhibitors”. I don’t understand the use of the word “sufficient” in this sentence.

Line 91 states that “Thirty-eight strains of Burkholderia spp. were tested”. However, table S2 only lists 36 strains of Burkholderia spp (plus 2 strains of Caballeronia zhejiangensis). What was the rationale for including two strains of Caballeronia zhejiangensis? See also line 144 and line 227/8.

Secion 2.6 states that the LOD was determined with Burkholderia cenocepacia AU1054. This seems to conflict with the abstract which provides LOD information for Burkholderia cenocepacia J2315 (line 23). These are listed as distinct strains according to Table S2.

Line 172: Can the authors prove a justification (or a reference) for using TSA at 1/10 strength?

Section 3.1. The sequence of the probe should be provided to allow others the opportunity to repeat the experimental work.

Line 203: I do not understand “……2151.7 cells/mL per 10 CFU/mL”. Please amend or explain.

Author Response

Thank you for your kind comments on the novelty of our study. 

Reviewer 2 Report

Comments to the Authors

The manuscript describes a flow cytometry method for specific detection and enumeration of BCC using a fluorescence-labeled oligo-nucleotide probe. This method has a great potential be to more sensitive alternative to culture-based methods to detect BCC in non-sterilized pharma-ceutical raw materials and products with regards to water-based environments. Overall the study reads well. Unfortunately, the manuscript needs to be thoroughly revised before it will be acceptable for publication. Some specific comments can be found below.

Some concerns and suggestion are as follows:

  1. Authors declared that“The average cell count using the Kef probe assay … were 25 similar to that of total plate count (TPC).”However, according to Section 2.3, “TPC” in this study was only a flow cytometry-based system RAPID-B, not traditional plating counting method. Thus, “TPC” in the manuscript needs to be revised.
  2. The title shows that the Kef probe assay could enumerate BCC. Unfortunately, the results of this study could not prove this point. First, if the authors use RAPID-B system as a reference standard, the linear relation (Section 3.4) between “cells/mL” and “CFU/mL” would be invalid. If the traditional plating counting method was used as a reference standard, the CFU number by plating counting and cells number by flow cytometry of BCC seemed so different, which did not comply with common sense (1 CFU ≈ 1 cell).
  3. How did the authors prepare the dilutions (10 and 102 CFU/mL) of BCC suspension? And how did the authors determine the precise CFU number of the dilutions? Why were the CFU number and cells number of BCC so different?
  4. Authors declared that the detection limits of the Kef probe assay was 1 CFU/mL. In consideration of the huge difference between “cells/mL” and “CFU/mL”, it is suggested to re-evaluate the detection limits carefully.
  5. It is necessary for authors to compare the Kef probe assay with other methods in discussion, concentrating on detection time, detection limit, cost, viable/dead cells recognition, and so on.
  6. Line 132, what is the purpose of “hybridization buffer”?
  7. The B. cenocepacia AU1054 strain was used to evaluate the detection limit in Section 2.6. Why was it not utilized in antiseptic solutions and surface water in Section 2.7?
  8. Figure 1b, the results of “TPC” and “65℃” could not be distinguished effectively. Please revise this figure.

Author Response

(The authors gave the same response as above.)

Round 2

Reviewer 1 Report

One of the recommendations has not been adequately addressed. It is important to highlight any limitations of the study. The authors should state that a limitation of the study is that not all species of BCC were included. Those that were not tested (including B. multivorans and B. vietnamiensis) should be listed.

Author Response

We thank the reviewer for the appreciative comments and constructive suggestions. To reflect the recommendation, we have added in Discussion “Despite the fact that our probe design covers all BCC strains, we were unable to test two majors strains encountered in cystic fibrosis patient B. multivorans, and B. vietnamiensis, due to a stock shortage.” (line 426-428).
